# Self-identity explains better breastfeeding intention of ethnic pregnant mothers of Western Nepal: Extending the theory of planned behavior

Chiranjivi Adhikari[1,2]*, Rojana Dhakal[1], Kapil Giri[1], Biddhya Bhandari[1], Rameshwor Baral[1], Krishna Prasad Pathak[3], Lal Bahadur Kunwar[4], Poshan Thapa[5], Yadu Ram Upreti[6], Khem Narayan Pokharel[7], Chhabi Lal Ranabhat[8]

1 School of Health and Allied Sciences, Pokhara University, Pokhara, Kaski, Nepal, 2 Indian Institute of Public Health Gandhinagar (IIPHG), Gandhinagar, India, 3 Department of Preventive Medicine, Federal University of São Paulo (UNIFESP), Sao Paulo, Brazil, 4 Medic Mobile, Asia, Kathmandu, Nepal, 5 School of Population Health, University of New South Wales, Sydney, Australia, 6 Central Department of Health and Physical Education, Tribhuvan University, Kathmandu, Nepal, 7 Development and Research Service International, Khumaltar, Lalitpur, Nepal, 8 Department of Health Promotion and Administration, Eastern Kentucky University, Richmond, Kentucky, United States of America

* chiranadhikari@gmail.com

**Data Availability Statement:** All data are in the manuscript and Supporting information files.

## Abstract

### Introduction

Breastfeeding intention is one of the strongest predictors of breastfeeding behavior and practice. The Theory of Planned Behavior (TPB), with its main construct, behavioral intention, is useful to predict actual behavior. While the literature has examined the implications of other theoretical notions such as self-identity, moral norm, descriptive norm, and socio-demographic variables, their roles remain unclear. Similarly, research on ethnic and low-income mothers is even insufficient. Therefore, given the original TPB constructs, our goal was to examine the role of extra theoretical constructs and specific demographic variables, and observe whether the original model would alter.

### Methods

A cross-sectional analytical study was carried out among 325 pregnant mothers in six purposively selected health care facilities of Kaski and Tanahu districts, using structured pro forma, from December 2018 to November 2019. We developed stepwise multivariate logistic regression from the entered and cleaned data, observed the effects on breastfeeding intention (BFI), and checked against multiple parameters. We interpreted the model with adjusted odds ratios and β coefficients, along with the variance explained.

### Results and discussions

Out of 325 pregnant women, more than half (54.8%) were primiparous, and the mean age was 25.1±5 years. All three theoretical constructs of TPB regressed the BFI significantly,

**Funding:** CA received a partial grant from School of Health and Allied Sciences (SHAS) of Pokhara University under the faculty grant no. 650-2074/ 2075. Url: https://pushas.edu.np/ The funders had no role in study design, data collection and analysis, decision to publish, or preparation of the manuscript.

**Competing interests:** CA and RD are faculty members under SHAS. This does not alter our adherence to PLOS ONE policies on sharing data and materials.

with 10.7%, the breastfeeding attitude dominating ($\beta$ = 0.734, p = .003), and the other two constructs—perceived breastfeeding control ($\beta$ = 0.659, p = .011) and breastfeeding subjective norm ($\beta$ = 0.504, p = .045). Interestingly, breastfeeding self-identity added further variance of 5.2% ($\beta$ = 0.955, p < .001), followed by income, with 3.2% ($\beta$ = 0.856, p = .001), both of which also tallied large effect sizes. The model parameters showed consistency and robustness. We discuss the variables of the theoretical and extended model for BFI as well as the arguments for the explained variance.

## Conclusion

Self-identity is a strong and independent predictor of breastfeeding intention and, therefore, should be considered to better prepare for breastfeeding, especially among ethnic pregnant mothers. The direct and intermediate effects of self-concept and income need further study with more robust designs.

## Introduction

Breastfeeding (BF) promotion is the most cost-effective intervention to advance mother-child health [1]. BF, including early initiation of it (EIBF), not only reduces infant and under-five mortalities but also plays key roles in maternal health promotion by protecting mothers against postpartum bleeding, breast and ovarian cancer, and osteoporosis [2,3]. So, breastfeeding should begin as early as possible and be maintained for as long as feasible. Breastfeeding intention (BFI), or intended to breastfeed, is a mother's plan before delivery to only breastfeed or to breastfeed and use formula [4]. It is an immediate precursor of breastfeeding behavior [5], and is closer to the conative or motivational aspect of BF attitude [6], which strongly predicts BF initiation [7] and related behaviors [7–9]. Since BF among primates, including humans, is, by and large, a learning and adaptive process [10], BFI is crucial and one of the strongest predictors of BF initiation [7] and the BF behavior change process. Having BFI independently, even formula-fed infants have similar health outcomes as breastfed infants [11].

Importantly, mothers across all parities, mainly nulliparous, do not adhere to BF, mainly EIBF [12]. Direct effects of theoretical constructs of the Theory of Planned Behavior (TPB) like behavioral attitude (BA), subjective norm (SN), and perceived behavioral control (PBC) have been found with varying effects on BFI, breastfeeding behavior (BFB), and its duration [7–9,11–15]. To adding up, the literature revealed a mixed effect of age [16,17]; income, employability, or deprivation [13,16–19]; race or ethnicity [7,18–22]; parity [12,17]; self-identity or motherhood identity [16,17,23,24]; moral norm [16,17]; knowledge [17,23,25]; health beliefs [25] and descriptive norm [16] with BFI and BF behaviors. Although knowledge showed no role in behavioral intention [26], earlier scientists had claimed it as at least a necessary component for a behavioral outcome [27].

In this case, with a paucity of studies carried out in Nepal, only 80% of newborn babies were found to be breastfed within one hour of delivery [28] whereas a cohort study showed an average intended BF of 28 months with almost universal BF, with 80% having positive SN (husband, mother, or mother-in-law preferred BF) [29]. Similarly, a clinic-based study of mothers with gestational diabetes mellitus (GDM) showed that knowledge, health beliefs, and an immediate family history of DM were found to be BFI predictors [25]. To further clarify,

we aimed to explain the role of knowledge, self-identity, moral norm, descriptive norm, and some socio-economic variables in the original theory of TPB in BFI.

## Materials and methods

### Study design, sampling and participants

We conducted a cross-sectional facility-based study visiting different six health facilities (Kaski-2, Tanahu-4) in Gandaki province between December 2018 and November 2019. We selected the health facilities purposively considering the attendance of ethnic pregnant mothers, and thereafter conveniently enrolled among them those who had completed their first trimesters. The sample size was calculated using Cochran's formula as $Z^2_{1-\alpha/2}\, pq/d^2$, with $Z_{1-\alpha/2}$ as the standard normal variate valued at 1.96, p as 29.8%, with a high intent to exclusively breastfeed, as reported by Odisha State of India [30], and the allowable error (d) as 0.05, the sample size was determined as 321. However, we enrolled 325 ethnic pregnant mothers.

### Scoring of constructs

We assessed breastfeeding attitude (BFA) with 12 statements, breastfeeding subjective norm (BFSN) with two, perceived breastfeeding control (PBFC) with three, breastfeeding intention (BFI) with five, breastfeeding self-identity (BFSI) with four, breastfeeding knowledge (BFK) with seven, and breastfeeding descriptive norm (BFDN) and breastfeeding moral norm (BFMN) each with three statements. All the statements were assessed on 5-point Likert scales, except for BFK and BFDN, with possible value ranges of 12–60, 2–10, 3–15, 5–25, 4–20, and 3–15 for BFA, BFSN, PBFC, BFI, BFSI, and BFMN, respectively. BFK had seven statements with value ranges of 0–1 and 1–3, whereas BFDN had three statements with value ranges of 0–3, 1–3, and 1–5. Possible composite value ranges for both BFK and BFDN were 2–11. We dummied the variables with median cut-offs for modeling (S1 Table 1–16 in S1 File).

### Data management

Coordinating with the staff of relevant health facilities, we collected data after the completion of antenatal check-ups using a pre-tested pro forma with face-to-face interviews in the Nepali language. We entered data in Epi-Data and imported it to SPSS to analyze. The data were refined with corrections for errors such as missing data, coding errors, and entry errors. For descriptive statistics, we used frequency and percentage. Cross tabulation for categorical data and dichotomized dummy variables was carried out with a chi-square test, and the variables with p≤.06 were carried over for stepwise logistic regression, starting with the theoretical predictors of TPB. We reported the adjusted odds ratio (aOR), along with beta coefficients and the model parameters. Before performing the final model, we carried out the Durbin-Watson test and variance inflation factor (VIF) with tolerance statistics to check the independence of errors and multicollinearity of the independent variables (Tolerance and VIF, respectively; BFA, 0.917, 1.091; BFSN, 0.899, 1.113; PBFC, 0.956, 1.046; BFDN, 0.889, 1.125; BFMN, 0.935, 1.070; BFSI, 0.953, 1.049; parity, 0.950, 1.053; and income, 0.973, 1.028) and revealed no multicollinearity. Parameters such as Negelkarke, and Cox and Snell Pseudo R squares, the Log likelihoods, and Akaike Information Criteria (AIC) were performed for model fit and to determine the variance of independent variables that explained the BFI model. Nepalese currency was transformed into US dollars at a rate of 110.7 as of the mid-period of the data collection [31].

## Ethical consideration

We obtained ethical approval from the Pokhara University Research Center, Institutional Review Committee (PURC, IRC, reference no. 35/076/077). We obtained the written consent of the participants along with the ethical aspects mentioned in the Helsinki Declaration—2013 version. Consent from the legal guardians and assent from adolescent mothers were obtained in cases below 18 years of age. In addition, written permissions from the health facilities were also obtained. A short breastfeeding counseling was given to mothers with a lower BFI and an unfavorable BF attitude.

## Results

### Descriptive statistics

Almost three-fourths (74.5%) of all mothers possessed Hinduism, and comparatively, husbands had higher educational status than their counterparts. Females were mostly engaged in household work, whereas husbands had foreign jobs as a major occupation (Table 1).

**Table 1. Socio-demographic characteristics of pregnant mothers.**

| Variables | Frequency(n = 325) | Percent (%) |
|---|---|---|
| **Age in years** | | |
| <20 | 43 | 13.2 |
| 20–30 | 243 | 74.8 |
| >30 | 39 | 12.0 |
| Mean ±SD (Min, Max) | 25.1±5.0 (16, 43) | |
| $^{\$}$**Family types** | | |
| Joint | 195 | 60.0 |
| Nuclear | 130 | 40.0 |
| **Religion** | | |
| Hindu | 242 | 74.5 |
| Buddhist | 36 | 11.1 |
| Christian | 25 | 7.7 |
| Muslim | 22 | 6.7 |
| **Ethnicity** | | |
| Dalit | 97 | 29.8 |
| Magar | 76 | 23.4 |
| Gurung | 59 | 18.1 |
| Newar | 25 | 7.7 |
| Muslim | 22 | 6.8 |
| Tamang | 20 | 6.2 |
| Others (Kumal, Limbu, Thakali, and Chaudhary) | 26 | 8.0 |
| **Parity** | | |
| Primiparous | 178 | 54.8 |
| Multiparous | 147 | 45.2 |
| **Educational level of mothers** | | |
| *Basic level and below | 114 | 35.1 |
| Secondary level | 178 | 54.7 |
| Higher education | 33 | 10.2 |
| **Educational status of husbands** | | |

*(Continued)*

**Table 1.** (Continued)

| Variables | Frequency(n = 325) | Percent (%) |
|---|---|---|
| *Basic level and below | 85 | 26.1 |
| Secondary level | 190 | 58.5 |
| Higher education | 50 | 15.4 |
| **Mother's occupation** | | |
| House-maker | 239 | 73.5 |
| Business# | 35 | 10.8 |
| Agriculture and labour | 28 | 8.6 |
| Services | 23 | 7.1 |
| **Husband's occupation** | | |
| Foreign employment | 99 | 30.5 |
| Services | 74 | 22.7 |
| Labour | 70 | 21.5 |
| Business | 59 | 18.2 |
| Agriculture | 23 | 7.1 |
| **Annual income (in US$)@** | | |
| Low (<3250) | 122 | 37.5 |
| High (= >3250) | 203 | 62.5 |
| Median ($Q_1$~$Q_3$) [Min~Max] | 3250 (2166~4332) [542~16246] | |

$Nuclear, parents and their children only; Joint, parents, their children and at least an additional member, such as grand-child or child's spouse or any third-generation offspring;

*Includes literates and illiterates;

#Includes 1 foreign employment;

@1 US$ = 110.7 NPR.

Scores of independent and dependent variables were found to be distributed between the range of 40–60% when merged into a dichotomy, except for PBC, which was two-thirds and one-thirds (67 vs. 33%) (Table 2).

## Bivariate and multivariate logistic regression modeling

Two original and one extended theoretical constructs—favorable breastfeeding attitude (BFA) (uOR (95% CI), 2.63 (1.67–4.14)), good perceived breastfeeding control (PBFC) (2.07 (1.30–3.32)), and favorable breastfeeding-related self-identity (BFSI) (2.17 (1.38–3.42))—as well as two socio-demographic variables—high-income (2.26 (1.43–3.59)) and multi-parity (1.61 (1.03–2.52)) were found to be associated (p's < .05) with breastfeeding intention (BFI) in bivariate analysis. Surprisingly, the original construct of the theory—compatible breastfeeding subjective norm (BFSN)—was only found marginally significant (uOR (95% CI), 1.53 (0.98–2.39); p, .060). Similarly, different socio-demographic variables like age, religion, husbands' and mothers' educational statuses and occupations, and family types were nonsignificant (p's>.05). Interestingly, although Gurung, Thakali, and Newar are regarded as ethnicities with a high human development index (HDI) compared with others, we did not observe a significant association between ethnicity and BFI (p = .426). In the same vein, extended theoretical constructs like breastfeeding knowledge (BFK), breastfeeding descriptive norm (BFDN), and breastfeeding moral norm (BFMN) were also observed to be nonsignificant (p's>.05) (Table 3).

**Table 2. Descriptive statistics of original and extended theoretical constructs.**

| Variables | Frequency(n = 325) | % |
|---|---|---|
| **Breastfeeding attitude (BFA)** | | |
| Unfavourable (<41) | 143 | 44.0 |
| Favourable (≥41) | 182 | 56.0 |
| Median ($Q_1$~$Q_3$) [Min~Max] | 41 (38~43) [29~54] | |
| **Breastfeeding subjective norm (BFSN)** | | |
| Compatible (pro-peer) (≥10) | 167 | 51.4 |
| Non-compatible (anti-peer) (<10) | 158 | 48.6 |
| Median ($Q_1$~$Q_3$) [Min~Max] | 10 (8~10) [4~10] | |
| **Perceived breastfeeding control (PBFC)** | | |
| Good (≥11) | 217 | 66.8 |
| Poor (<11) | 108 | 33.2 |
| Median ($Q_1$~$Q_3$) [Min~Max] | 11 (9~11) [4~12] | |
| **Breastfeeding intenders (BFI)** | | |
| Good intenders (≥24) | 192 | 59.1 |
| Poor intenders (<24) | 133 | 40.9 |
| Median (Q1~Q3) [Min~Max] | 24 (21~25) [15~25] | |
| **Breastfeeding self-identity (BFSI)** | | |
| More flexible (pro-social) (≥14) | 190 | 58.5 |
| Less flexible (pro-self) (<14) | 135 | 41.5 |
| Median ($Q_1$~$Q_3$) [Min~Max] | 14 (13~16) [7~20] | |
| **Breastfeeding knowledge (BFK)** | | |
| Good knowledge (≥3) | 197 | 60.6 |
| Poor knowledge (<3) | 128 | 39.4 |
| Median (Q1~Q3) [Min~Max] | 3 (1~5) [0~7] | |
| **Breastfeeding descriptive norms (BFDN)** | | |
| Compatible (pro-siblings) (≥10) | 175 | 53.8 |
| Non-compatible (anti-siblings) (<10) | 150 | 46.2 |
| Median (Q1~Q3) [Min~Max] | 10 (9~11) [4~12] | |
| **Breastfeeding moral norms (BFMN)** | | |
| Compatible (pro-moral) (≥10) | 191 | 58.8 |
| Non-compatible (anti-moral) (<10) | 134 | 41.2 |
| Median (Q1~Q3) [Min~Max] | 16 (14~16) [8~20] | |

Breastfeeding intention (BFI) as an outcome variable, and the explanatory variables, the BFA, the PBFC (both p's < .05), and the BFSN (p = .060), the original constructs of the theory, were included as an initial set of variables in multivariate modeling, further adjusting the parity, income, and the BFSI in the second, third, and final steps consecutively (Table 4, Fig 1).

In the final step, along with three original constructs of TPB, including the BFSN, an extended construct–BFSI, and a socio-economic variable–income, significantly explained the BFI. Household income and BFSI increased the variance of the model by 3.2 and 5.2 percent points, respectively, nearly equal additional variance (Cox and Snell pseudo $R^2$, from 7.9 to 15.1%; and similar proportion of Negelkarke pseudo $R^2$). In addition, these extended variables showed significance with aORs (95% CI) of 2.35 (1.42–3.92) and 2.60 (1.57–4.30), respectively. Although parity was associated with BFI (p = .025, Table 3) in the initial model, it was nonsignificant (p>0.05) in the final step. When Cox and Snell, and Negelkarke pseudo $R^2$'s, both

**Table 3. Socio-demographic associative characteristics of breastfeeding intention.**

| Variables | Breastfeeding intenders | | $x^2$-statistic (p-value) | uOR (95%CI) |
|---|---|---|---|---|
| | Poor, n = 133 (%) | Good, n = 192 (%) | | |
| **Age (Years)** | | | | |
| < 20 | 19 (14.3) | 24 (12.5) | 0.60 (0.740) | |
| 20–30 | 100 (75.2) | 143 (74.5) | | |
| > 30 | 14 (10.5) | 25 (13.0) | | |
| **Parity** | | | | |
| Primi | 82 (61.7) | 96 (50.0) | 4.31 (0.025) | Ref |
| Multi | 51 (38.3) | 96 (50.0) | | 1.61 (1.03–2.52) |
| **Annual income** | | | | |
| Low | 65 (48.9) | 57 (29.7) | 12.33 (<0.001) | Ref |
| High | 68 (51.1) | 135 (70.3) | | 2.26 (1.43–3.59) |
| **Janajatis (ethnicity)** | | | | |
| Advantaged# | 36 (27.1) | 49 (25.5) | 0.10 (0.426) | |
| Disadvantaged@ | 97 (72.9) | 143 (74.5) | | |
| **Religion** | | | | |
| Hindu | 96 (72.2) | 146 (76.0) | 1.68 (0.643) | |
| Buddhist | 14 (10.5) | 22 (11.5) | | |
| Christian | 13 (9.8) | 12 (6.2) | | |
| Muslim | 10 (7.5) | 12 (6.2) | | |
| **Mother's education** | | | | |
| Below secondary | 51 (38.3) | 63 (32.8) | 1.06 (0.181) | |
| Secondary and above | 82 (61.7) | 129 (67.2) | | |
| **Husband's education** | | | | |
| Below secondary | 36 (27.1) | 49 (25.5) | 0.10 (0.426) | |
| Secondary and above | 97 (72.9) | 413 (74.5) | | |
| $^\$$**Family types** | | | | |
| Nuclear | 58 (43.6) | 72 (37.5) | 1.22 (0.161) | |
| Joint | 75 (56.4) | 120 (62.5) | | |
| **Mothers' occupation** | | | | |
| Business | 15 (11.3) | 19 (9.9) | 2.65 (0.449) | |
| Service | 7 (5.3) | 16 (8.3) | | |
| Housewife | 96 (72.2) | 143 (74.5) | | |
| Others* | 15 (11.3) | 14 (7.3) | | |
| **Husband occupation** | | | | |
| Agriculture | 12 (9.0) | 11 (5.7) | 3.15 (0.534) | |
| Business | 24 (18.0) | 35 (18.2) | | |
| Service | 30 (22.6) | 44 (22.9) | | |
| Labour | 32 (24.1) | 38 (19.8) | | |
| Foreign employment | 35 (26.3) | 64 (33.3) | | |
| **Breastfeeding knowledge (BFK)** | | | | |
| Poor knowledge | 48 (36.1) | 80 (41.7) | 1.02 (0.185) | |
| Good knowledge | 85 (63.9) | 112 (58.3) | | |
| **Breastfeeding descriptive norm (BFDN)** | | | | |
| Non-compatible | 65 (48.9) | 85 (44.3) | 0.67 (0.240) | |
| Compatible | 68 (51.1) | 107 (55.7) | | |
| **Breastfeeding moral norm (BFMN)** | | | | |

*(Continued)*

**Table 3.** (Continued)

| Variables | Breastfeeding intenders | | x²-statistic (p-value) | uOR (95%CI) |
|---|---|---|---|---|
| | **Poor, n = 133 (%)** | **Good, n = 192 (%)** | | |
| Non-compatible | 58 (43.6) | 76 (39.6) | 0.53 (0.271) | |
| Compatible | 75 (56.4) | 116 (60.4) | | |
| **Breastfeeding-related self-identity (BFSI)** | | | | |
| Unfavorable | 70 (52.6) | 65 (33.9) | 11.41 (0.001) | Ref |
| Favorable | 63 (47.4) | 127 (66.1) | | 2.17 (1.38–3.42) |
| **Breastfeeding attitude (BFA)** | | | | |
| Unfavorable | 77 (57.9) | 66 (34.4) | 17.64 (<0.001) | Ref |
| Favorable | 56 (42.1) | 126 (65.6) | | 2.63 (1.67–4.14) |
| **Breastfeeding subjective norm (BFSN)** | | | | |
| Non-compatible | 73 (54.9) | 85 (44.3) | 3.55 (0.060) | Ref |
| Compatible | 60 (45.1) | 107 (55.7) | | 1.53 (0.98–2.39) |
| **Perceived breastfeeding control (PBFC)** | | | | |
| Poor | 57 (42.9) | 51 (26.6) | 9.40 (0.002) | Ref |
| Good | 76 (57.1) | 141 (73.4) | | 2.07 (1.30–3.32) |

$^\$$Nuclear, parents and their children only; Joint, parents, their children and at least an additional member, such as grand-child or child's spouse or any third-generation offspring;

*Agriculture, foreign job and laborer;

#Newar, Thakli and Gurung;

@Dalit, Magar, Muslim, Tamang, Kumal, Rai, Limbu and Chaudhary.

found to be increasing, the -2 log likelihood and the Akaike Information Criteria (AIC), both decreasing and predicting a better fit of the model, as it progressed (Table 4, Fig 1).

The final model showed that BFSI ($\beta$ = 0.955; p < .05) and income ($\beta$ = 0.856; p = .001) dominated with large effect sizes, whereas the coefficients of other covariates remained between medium and large ($\beta$'s, 0.39–0.73, with p's < .05 except for parity (p = .12) (Table 5).

## Discussion

We observed the effects of an extended theoretical construct–breastfeeding social identity (BFSI)—and a socioeconomic variable—income, on breastfeeding intention (BFI), in addition to the original constructs of the theory of planned behavior (TPB)—breastfeeding attitude (BFA), breastfeeding subjective norm (BFSN), and perceived breastfeeding control (PBFC). The total variance accounted for by these two variables was nearly equal to that of the originals ($\Delta R^2$, from 10.7 to 20.3%), and the larger contribution ($\Delta R^2$, 5.2%) was brought about by breastfeeding self-identity (BFSI). Although parity was found to affect in the earliest model, its adjusted effect was nullified in the subsequent models. Herein, we discuss in what ways parity affects the BFI. We also discuss breastfeeding knowledge (BFK), although it failed to meet statistical criteria (p>.060) for model entry, and parity, which remained nonsignificant in the final model. In addition, we further discuss, the role of self-concept, as it was found with highly networked predictors, implicating the BFI (Fig 2).

A longitudinal study among the Iranian mothers revealed that only two original constructs —BFA and PBFC—explained 23% of the variance in BFI, whereas BFSN was found ineligible to enter the model [12]. Another study carried out in Bangladesh showed that one-fourth (25%) variance in BFI was only brought about by BFA, with only 2% further added by BFSN

**Table 4. Multivariate models with stepwise adjusted effects of predictors.**

| #Variables | Breast Feeding Intenders | | $x^2$-statistic (p-value) in step1 | Step 1 uOR (95%CI) | Step 2 aOR (95% CI) | Step 3 aOR (95% CI) | Step 4 aOR (95%CI) |
|---|---|---|---|---|---|---|---|
| | Poor n = 133 (%) | Good n = 192 (%) | | | | | |
| **Breastfeeding attitude (BFA)** | | | | | | | |
| Unfavorable | 77 (57.9) | 66 (34.4) | 17.64 (<0.001)** | Ref | | | |
| Favorable | 56 (42.1) | 126 (65.6) | | 2.48 (1.56–3.94) | 2.42 (1.52–3.85) | 2.11 (1.31–3.41) | 2.08 (1.28–3.40) |
| **Breastfeeding subjective norm (BFSN)** | | | | | | | |
| Noncompatible | 73 (54.9) | 85 (44.3) | 3.55 (0.060) | Ref | | | |
| Compatible | 60 (45.1) | 107 (55.7) | | 1.37 (0.86–2.19) | 1.39 (0.847–2.22) | 1.47 (0.92–2.37) | 1.66 (1.01–2.71) |
| **Perceived breastfeeding control (PBFC)** | | | | | | | |
| Poor | 57 (42.9) | 51 (26.6) | 9.40 (0.002)** | Ref | | | |
| Good | 76 (57.1) | 141 (73.4) | | 1.89 (1.16–3.07) | 1.88 (1.15–3.05) | 1.82 (1.11–2.98) | 1.93 (1.16–3.22) |
| **Parity** | | | | | | | |
| Primi | 82 (61.7) | 96 (50.0) | 4.30 (.037)* | | Ref | | |
| Multi | 51 (38.3) | 96 (50.0) | | | 1.5 (0.95–2.43) | 1.66 (1.03–2.69) | 1.48 (0.90–2.42) |
| **Income** | | | | | | | |
| Low | 65 (48.9) | 57 (29.7) | 12.33 (< .001)** | | | Ref | |
| High | 68 (51.1) | 135 (70.3) | | | | 2.09(1.28–3.42) | 2.35 (1.42–3.92) |
| **BreastfeB Breastfeeding self -identity (BFSI)** | | | | | | | |
| Unfavorable | 70 (52.6) | 65 (33.9) | 11.41 (0.001)** | | | | Ref |
| Favorable | 63 (47.4) | 127 (66.1) | | | | | 2.60 (1.57–4.30) |
| **Model summary** | | | | | | | |
| Negelkarke Pseudo $R^2$ (%) | | | | 10.7 | 11.9 | 15.1 | 20.3 |
| $\Delta R^2$ (%) | | | | 10.7 | 1.2 | 3.2 | 5.2 |
| Cox and Snell Pseudo $R^2$ (%) | | | | 7.9 | 8.8 | 11.2 | 15.1 |
| -2 Log likelihood | | | | 412.8 | 409.8 | 401.1 | 386.8 |
| $AIC | | | | 420.8 | 419.8 | 413.1 | 400.8 |

[#]DW statistic = 1.56; Tolerance, Min-Max, 0.89–0.97; VIF Min-Max, 1.03–1.13;

[$]AIC, Akaike Information Criteria.

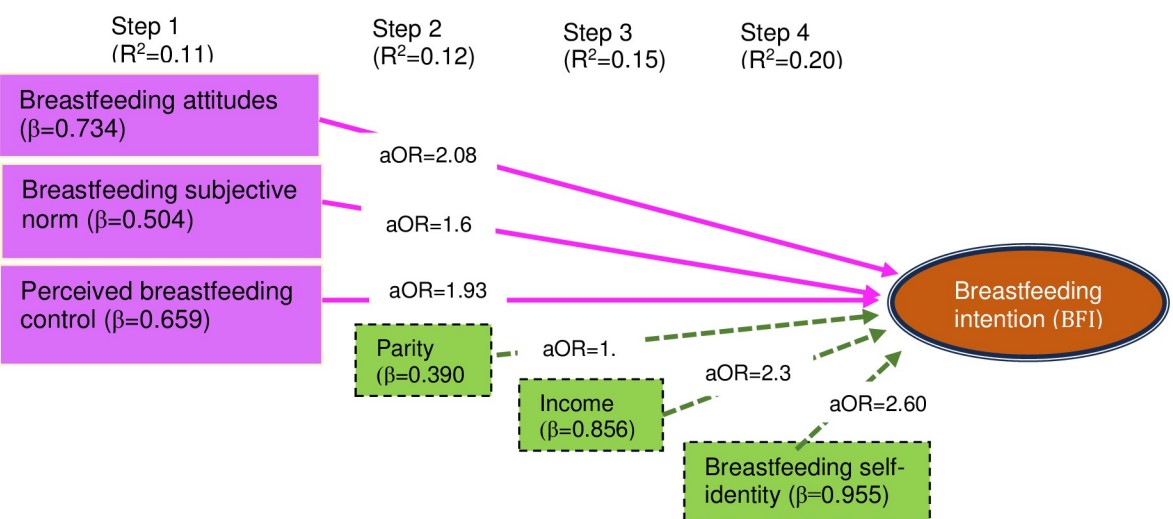

**Fig 1. Step-wise input of regressors, their effect sizes (in boxes) (β and aOR) in final model, and progressive variances (in parentheses, with steps); pink boxes and solid lines indicate original constructs; green boxes and broken lines indicate extended variables; all variables were significant (p's < .05) except parity (p = .122).**

**Table 5. Beta coefficients and related parameters of the final model.**

| Covariate | β | SE | Wald | Df | P-value |
|---|---|---|---|---|---|
| BFA | 0.734 | 0.250 | 8.617 | 1 | .003 |
| BFSN | 0.504 | 0.251 | 4.030 | 1 | .045 |
| PBFC | 0.659 | 0.260 | 6.438 | 1 | .011 |
| Parity | 0.390 | 0.252 | 2.397 | 1 | .122 |
| Income | 0.856 | 0.260 | 10.864 | 1 | .001 |
| BFSI | 0.955 | 0.257 | 13.816 | 1 | < .001 |
| Constant | -3.197 | 0.[658 | 23.609 | 1 | < .001 |

[32]. Likewise, in a similar study carried out among the mothers of economic hardship, all original constructs explained 44% ($\Delta R^2 = 0.44$) [16]. Although quite less than these studies, our model predicted only 10.7% of total intention, including the BFSN, showed the likely trait (marginally significant; p = .060; Table 3), so we included it in the model with further adjustment, and it was significant (aOR, 1.66; 95% CI, 1.01–2.71; Table 4). It was only the ethnic mothers who were included in our study, the majority of them in primigravida, and during their second trimester, which might have shown less predictability as compared to these studies; further critical judgements are warranted though [14,16,32].

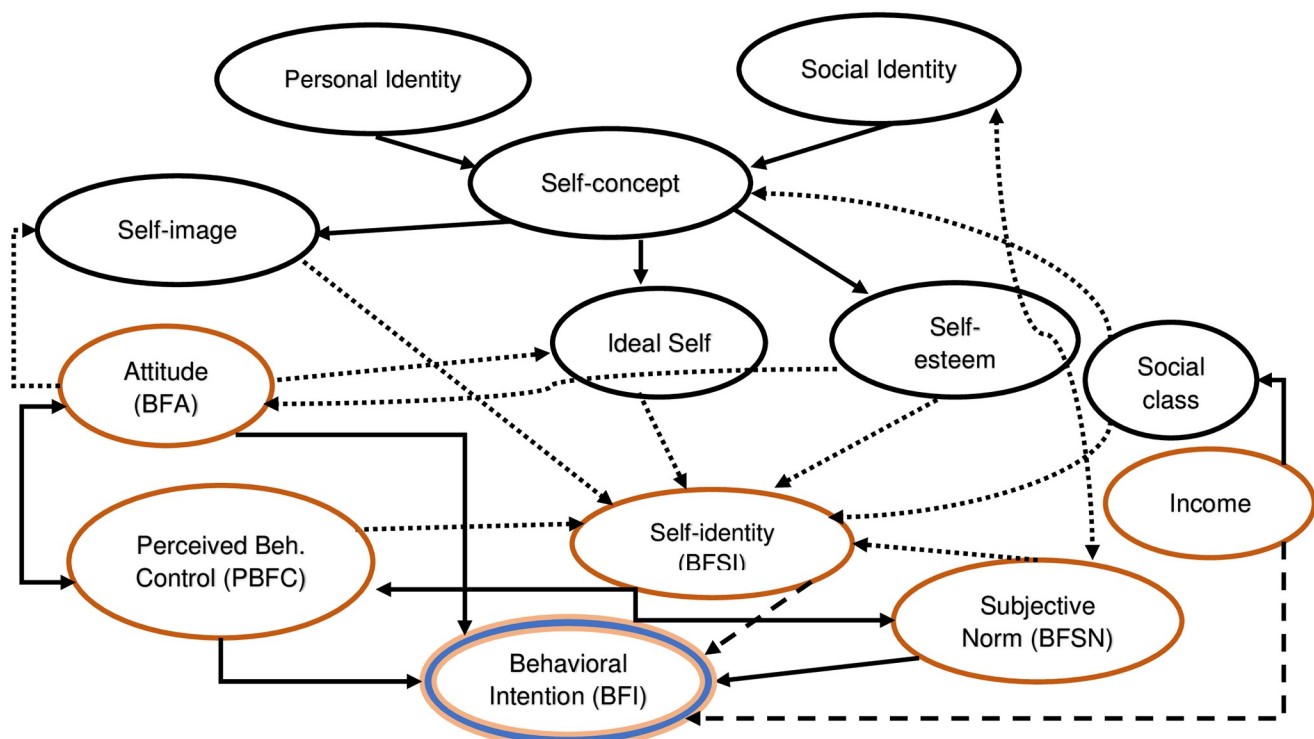

**Fig 2.** Concept map of BFI predictors (orange ovals-assessed from our study; black ovals- from literature) and their associations (unidirectional- single arrow-headed; bidirectional- double arrow-headed); dotted lines indicate plausible/probable (from literature), dashed lines indicate probable (from our study and literature both), and solid lines indicate established theoretical connections (from our study and/or literature); BFI = Breastfeeding intention (double-lined oval), BFA = Breastfeeding attitude, PBFC = Perceived breastfeeding control, BFSN = Breastfeeding subjective norm, BFSI = Breastfeeding self-identity.

Breastfeeding intention (BFI) was found to be influenced by three highly networked predictors: breastfeeding self-identity (BFSI), breastfeeding attitude (BFA), and self-concept (Fig 2), and also, showed high effects (Fig 1, Table 5). BFA and its direct causal links with the BFI have been observed in our study and also in reviews [33,34], and intermediary links, such as BFA–self-image–BFSI [35] and BFA–ideal self–BFSI [36], can be found discussed elsewhere. We focus our discussion mainly on self-concept, self-identity, or BFSI, and the direct and intermediary effects of income on BFI.

Self-concept is developed through a continuous process of the self, interacting with the social identity and has been explored as both observable and dispositional behaviors, like total outputs of BF (more distal) and in the abstract, like looking pretty or handsome (more proximal). In this context, destruction of the process may occur by lowering one's realistic success aspirations for such unjustifiable excuses being assigned by society, mostly to those at lower statuses of the social ladder, like among ethnicities, and eventually leading to low self-esteem [37]. Testing hypotheses of maternal self-concept associating BF showed not only the total self-concept, but its individual dimensions like reflecting self-satisfaction, behavior, moral worth, value as a family member, and physical appearance also showed significant regression coefficients [38].

Self-identity is a relatively enduring characteristic that people ascribe to themselves, which develops later as a form of socially given linguistic categorization [39] and is attributed as an extrinsic motivation [40]. Further, as applied in BF, a breastfeeding woman develops herself as her self-identity, characterizing herself as BSFI, working stronger with charitableness and not with just helpfulness [41]. This further makes her intend to breastfeed her baby [16,24,42]. Although breastfeeding was found to be challenged by strain and pain, and by role and gender conflicts, it still helps develop the BFSI with a favorable interplay of intrapersonal-biological, interpersonal-relational, sensual, and social elements that drive mothers to breastfeed, a postcursor of BFI [43]. However, when a BF mother also has other salient and pressing identities that attract media attention, such as an athlete, there may be an antagonizing effect, causing psychological distress and a feeling of guilt among the athlete BF mothers when they have to take time for training and competition [44].

Although income was found to be less networked, it showcased a strong predictor in our model (Table 5, Figs 1 and 2) and also in another study [20]. For this, we reviewed the covariates explaining the BFI among low-income mothers. We observed that higher education and knowledge [19,20,43,44], longer BF experience and self-efficacy [19,44], obtaining husband, family, and social support (both perceived and actual) [15,20,45,46], having lesser parities and fewer children [20,45], ethnicities and races, further explaining more among the Black than Hispanic/White, more among Afro-Caribbean than African-American [15,21], and immigrants than aboriginal [15,21,45], were found to be of higher intents of BF. Clearly, it may be hypothesized that among these factors, knowledge, experience, and education interplay for self-efficacy and further help develop confidence, which supports BF, whereas parity, social supports, ethnicity, and immigrant statuses interplay for self-concept and self-identity, mediated through and as a major component of social class, though further investigation is warranted.

Notably, the current study not only showed a nonsignificant association with BF knowledge (BFK) but the education levels of both mothers and husbands were also in the same vein. Conversely, an antenatal clinic-based study regarding GDM carried out in Nepal showed a small effect size ($\beta = 0.18$, $p = .020$). Another study carried out in Iran also showed a similar effect ($\beta = 0.103$, $p = < .01$) [18]. Clearly, the study of in-depth interviews with breastfeeding mothers revealed that not the theoretical BFK, but the embodied knowledge that is gained through direct observation or practical experience may enhance BF [23]. Further, aligning with this

finding, the significant effect of breast-self-examination and mammography knowledge, delivered through demonstration with a three-dimensional silicone dummy, on attitude was observed but not on subjective norm or behavioral control [47]. With such a three-dimensional dummy, the knowledge may be 'embodied', or, for that matter, the attitude may have changed. This shows that theoretical knowledge may induce a cognitive attitude or, to some extent, an affective one but not a strong intent. For better prediction and explanation of behavioral intention, the TPB should be integrated with motivation to perform the behavior, which may be explained further by Ajzen and Kruglanski's newer theory—the reasoned action in the service of goal pursuit [48].

Parity, with successive multiparity, is generally seen as an influencing factor of BFI [49]; however, the current study showed a nonsignificant beta value. Furthermore, a two-year longitudinal study showed that parity significantly interacts with the PBC, and only the PBC was found to be a significant predictor [50]. A large-scale retrospective cohort carried out in Canada also showed higher odds of BFI among older women without health problems who were cared for exclusively by midwives [51]. However, a multiparous mother who intends to breastfeed for 12 months, has a lesser probability of stopping BF before that than a primiparous mother [49]. So, the parity may influence BFI indirectly, or with an interaction effect; further studies are warranted.

## Limitations

There are some potential limitations to the study. Firstly, despite significant odds ratios and beta coefficients for the given predictors, the total variance explained in the final model is around one-fifth, which may be due to the dichotomized dummy variables of the ordinal scales and hence cautiously interpreted. Second, less frequent networked predictors such as self-esteem, social identity, ideal self, and self-image and their association with other theoretical constructs, such as BFSN and BFA, might be important in a multifactorial causal chain but are beyond the scope of this paper. Also, pregnancy-related experiences, such as happiness or unhappiness, might influence the BFI [19], which is beyond the scope of this study. Thirdly, the findings of purposively selected health care service points and the convenient samples from them should be cautiously inferred. Finally, some behavioral factors such as smoking and pre-pregnancy obesity, which are found to be associated with lower odds of exclusive BF [52], and extraneous factors such as traditional beliefs, including culture and trend, could not be addressed, so they may need further adjustment and exploration for interpretation.

## Conclusion

Breastfeeding self-identity (BFSI) is an additional independent and strong predictor of the BFI and may explain even better than each of the original TPB constructs—breastfeeding attitude (BFA), subjective norm (BFSN), and perceived control (PBFC)—among ethnic pregnant mothers, and so should be considered in preparing pregnant mothers for breastfeeding. Stronger study designs examining the direct and intermediate effects of self-concept and income among ethnic mothers are further necessary.

## Supporting information

**S1 File. S1 Table 1–16.** Constructs and measurement.
(DOCX)

**S1 Data.**
(SAV)

## Acknowledgments

We duly acknowledge former director of the School of Health and Allied Sciences (SHAS), Dr. Damaru Prasad Paneru, the executives of the SHAS Research Management Cell (SHAS-RMC), and the Institutional Review Committee of Pokhara University. We are greatly thankful to the MPH graduates of SHAS- Bibechana Sapkota, Kamal Bahadur Budha, and Bijaya Parajuli, for their support. We are also thankful to Mr. Shubham Sharma and Dr. VP Varna, MPH graduates of Indian Institute of Public Health-Gandhinagar (IIPHG), India, for their technical support.

## Author Contributions

**Conceptualization:** Chiranjivi Adhikari, Rojana Dhakal, Yadu Ram Upreti.

**Data curation:** Chiranjivi Adhikari, Kapil Giri, Biddhya Bhandari, Rameshwor Baral, Lal Bahadur Kunwar, Khem Narayan Pokharel.

**Formal analysis:** Chiranjivi Adhikari, Kapil Giri, Lal Bahadur Kunwar, Yadu Ram Upreti, Chhabi Lal Ranabhat.

**Funding acquisition:** Chiranjivi Adhikari, Kapil Giri, Biddhya Bhandari.

**Investigation:** Chiranjivi Adhikari, Kapil Giri, Biddhya Bhandari, Rameshwor Baral, Yadu Ram Upreti, Khem Narayan Pokharel, Chhabi Lal Ranabhat.

**Methodology:** Chiranjivi Adhikari, Rojana Dhakal, Krishna Prasad Pathak, Poshan Thapa, Yadu Ram Upreti, Khem Narayan Pokharel, Chhabi Lal Ranabhat.

**Project administration:** Chiranjivi Adhikari, Kapil Giri, Rameshwor Baral, Krishna Prasad Pathak, Lal Bahadur Kunwar.

**Resources:** Chiranjivi Adhikari, Kapil Giri, Rameshwor Baral, Krishna Prasad Pathak.

**Software:** Chiranjivi Adhikari, Kapil Giri.

**Supervision:** Chiranjivi Adhikari, Rojana Dhakal.

**Validation:** Chiranjivi Adhikari, Krishna Prasad Pathak, Poshan Thapa, Chhabi Lal Ranabhat.

**Visualization:** Chiranjivi Adhikari, Khem Narayan Pokharel.

**Writing – original draft:** Chiranjivi Adhikari, Rojana Dhakal, Kapil Giri, Biddhya Bhandari, Rameshwor Baral, Krishna Prasad Pathak, Lal Bahadur Kunwar, Poshan Thapa, Yadu Ram Upreti, Khem Narayan Pokharel, Chhabi Lal Ranabhat.

**Writing – review & editing:** Chiranjivi Adhikari, Rojana Dhakal, Kapil Giri, Biddhya Bhandari, Rameshwor Baral, Krishna Prasad Pathak, Lal Bahadur Kunwar, Poshan Thapa, Yadu Ram Upreti, Chhabi Lal Ranabhat.

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
