## [Decision Letter · Decision Letter 0]

19 Feb 2024

PONE-D-23-04076Self-identity explains better breastfeeding intention of ethnic pregnant mothers: Extending the theory of planned behaviorPLOS ONE

Dear Dr. Adhikari,

Thank you for submitting your manuscript to PLOS ONE. After careful consideration, we feel that it has merit but does not fully meet PLOS ONE’s publication criteria as it currently stands. Therefore, we invite you to submit a revised version of the manuscript that addresses the points raised during the review process.

**Please address the comments' raised by the reviewers annexed below and submit your revision. **

We look forward to receiving your revised manuscript.

Kind regards,

Abel Fekadu Dadi, Ph.D.

Academic Editor

PLOS ONE

Journal Requirements:

3. Thank you for providing the following Funding Statement:  

   "This research was conducted with partial financial support from School of Health and Allied Sciences (SHAS) of Pokhara University, Nepal, however, the authorities had no role in design, data collection, analysis and manuscript preparation. CA and RD are permanent faculty members of SHAS, and were benefitted for a week-long leave for data collection and study related management. KG, BB and RB were undergraduate research students of SHAS, who managed their time in this study, without being affected in their own research projects. The authors declare that there are not any other commercial or financial relationships that could be construed as a potential conflict of interest."

We note that one or more of the authors is affiliated with the funding organization, indicating the funder may have had some role in the design, data collection, analysis or preparation of your manuscript for publication; in other words, the funder played an indirect role through the participation of the co-authors. 

If the funding organization did not play a role in the study design, data collection and analysis, decision to publish, or preparation of the manuscript and only provided financial support in the form of authors' salaries and/or research materials, please review your statements relating to the author contributions, and ensure you have specifically and accurately indicated the role(s) that these authors had in your study in the Author Contributions section of the online submission form. Please make any necessary amendments directly within this section of the online submission form.  Please also update your Funding Statement to include the following statement: “The funder provided support in the form of salaries for authors [insert relevant initials], but did not have any additional role in the study design, data collection and analysis, decision to publish, or preparation of the manuscript. The specific roles of these authors are articulated in the ‘author contributions’ section.” 

If the funding organization did have an additional role, please state and explain that role within your Funding Statement. 

Please also provide an updated Competing Interests Statement declaring this commercial affiliation along with any other relevant declarations relating to employment, consultancy, patents, products in development, or marketed products, etc.  

Additional Editor Comments:

Please address reviewers' comment annexed below.

Reviewers' comments:

Reviewer's Responses to Questions

**Comments to the Author**

1. Is the manuscript technically sound, and do the data support the conclusions?

Reviewer #1: Partly

Reviewer #2: Yes

2. Has the statistical analysis been performed appropriately and rigorously? 

Reviewer #1: Yes

Reviewer #2: Yes

3. Have the authors made all data underlying the findings in their manuscript fully available?

Reviewer #1: Yes

Reviewer #2: Yes

4. Is the manuscript presented in an intelligible fashion and written in standard English?

Reviewer #1: Yes

Reviewer #2: No

5. Review Comments to the Author

Reviewer #1: Thank you for sending this paper. This paper reads well. Following points for further revision.

1. Title- this study was conducted in Nepal, reader should know from the title, and they start to assume the context. Currently, I knew it was about Nepal when I reached to mid of introduction section.

2. Brief context ( one sentence) is needed about the problem in the abstract section.

3. Readers are left to assume many thing in the introduction section- please elaborate it contextualizing breast feeding, and available literature in the topic in Nepal, culture, practices, and trends on it. Secondly, it is necessary to detailed in PTB, thirdly, research gaps, significance and objective. Elaborate it to at least two pages.

4. Methods section also needs more detail explanation- research context, selection of participants, variables, data analysis model, results section reported beta coefficient but that was not described in the methods section. Very importantly, there is no data how four constructs- BFA, PBFC, BFSI, and BFSN were created (as these were presented quantitatively).

5. Table can be included in the findings section- that reads easy with the text.

6. Discussion section lacks contextualities of findings relating with policy and programs of Nepal. Interpretation should be focused comparing with other studies in Nepal.

Reviewer #2: The document is informative and original.

It has an ethical statement clearly stated.

It is better if the document adds operational definitions and discusses all possible reasons for an association.

6. PLOS authors have the option to publish the peer review history of their article (what does this mean?). If published, this will include your full peer review and any attached files.

Reviewer #1: No

Reviewer #2: **Yes: **ESHETU YISIHAK UKUMO

---

## [Author Response · Author response to Decision Letter 0]

3 Apr 2024

Response to Editor

Q

 RE

Thank you! We reviewed main-text, funding statement, CoI, Acknowledgement and revised as per ‘Title’ and ‘Body formatting guidelines’ of the PLOS ONE. 

Q

Please note that funding information should not appear in any section or other areas of your manuscript. We will only publish funding information present in the Funding Statement section of the online submission form. Please remove any funding-related text from the manuscript.

RE

Funding information has been removed from the manuscript part. 

Q

We note that one or more of the authors is affiliated with the funding organization, indicating the funder may have had some role in the design, data collection, analysis or preparation of your manuscript for publication; in other words, the funder played an indirect role through the participation of the co-authors.

If the funding organization did not play a role in the study design, data collection and analysis, decision to publish, or preparation of the manuscript and only provided financial support in the form of authors' salaries and/or research materials, please review your statements relating to the author contributions, and ensure you have specifically and accurately indicated the role(s) that these authors had in your study in the Author Contributions section of the online submission form. Please make any necessary amendments directly within this section of the online submission form.

Please also update your Funding Statement to include the following statement: “The funder provided support in the form of salaries for authors [insert relevant initials], but did not have any additional role in the study design, data collection and analysis, decision to publish, or preparation of the manuscript. The specific roles of these authors are articulated in the ‘author contributions’ section.”

RE

▪Funding statement is revised as:

“This research was funded from School of Health and Allied Sciences (SHAS) of Pokhara University, Nepal (Faculty Research Grant no. 650-2074/2075), The funder provided a support in the form of paid leaves of two weeks for data collection [CA], but did not have any additional role in the study design, data analysis, decision to publish, or preparation of the manuscript. The specific roles of these authors are articulated in the ‘author contributions’ section.”

▪Author contribution revised as

Conceptualization (CA, YRU, RD, KG); Data Curation (CA, KG, BB, RB); Formal Analysis (CA, KG, CLR, LBK, PT, YRU);Fund Acquisition (CA); Investigation (CA, KG, BB, RB); Methodology (CA, RD, PT, CLR), Project Administration (CA, KG, RB); Resources (CA, KG, RB); Software (CA, KG, LBK); Supervision (RD, CA); Validation (CLR); Visualization (CA); Writing- Original Draft Preparation (All Authors); Writing- Review & Editing (CA, RD, CLR, PT) 

▪Funding organization’s role is included in the funding statement

▪Competing interest is updated as:

CA and RD are faculty members under SHAS. KG, BB and RB were undergraduate final semester research students under CA. This does not alter our adherence to PLOS ONE policies on sharing data and materials. 

Q

Please provide a complete Data Availability Statement in the submission form, ensuring you include all necessary access information or a reason for why you are unable to make your data freely accessible. If your research concerns only data provided within your submission, please write "All data are in the manuscript and/or supporting information files" as your

Data Availability Statement.

RE

▪Revised as “All data are in the manuscript and supporting information files.”

REVIEWER # 1

Q

Title- this study was conducted in Nepal, reader should know from the title, and they start to assume the context. Currently, I knew it was about Nepal when I reached to mid of introduction section.

RE

We are thankful to the reviewer. The title is revised as “Self-identity explains better breastfeeding intention of ethnic pregnant mothers of Western Nepal: Extending the theory of planned behavior”

Q

Brief context (one sentence) is needed about the problem in the abstract section.

RE

Sorrily I would like to reiterate that it has been mentioned as “…Although effects of additional theoretical constructs like self-identity, moral norm, and descriptive norm, and socio-demographic variables have been explored in the literature, their roles are not clear. Similarly, such exploration among the ethnic and low-income mothers are even scarce…” Kindly suggest further if not sufficient. 

Q

Readers are left to assume many things in the introduction section- please elaborate it contextualizing breast feeding, and available literature in the topic in Nepal, culture, practices, and trends on it. Secondly, it is necessary to detailed in PTB, thirdly, research gaps, significance and objective. Elaborate it to at least two pages

RE

▪Thank you for this comment, especially, the need of theoretical explanation of TPB, it is found elsewhere (Ajzen I. The theory of planned behavior. Organizational behavior and human decision processes. 1991 Dec 1;50(2):179-211.), including its evidence synthesis in breast feeding (https://link.springer.com/article/10.1007/s10995-018-2453-x), and BF testing in TPB rather than theoretical description, is our study aim. 

▪We agree that culture is obviously related to the constructs and practices of BF intention and behavior, however, this will be out of the scope of our study. Moreover, we are concerned only with cross-sectional empirical evidence, the trend, we assume, would also be out of the scope of this study. However, agreed these are important, we included in limitations: The trend and cultural influences …

▪PLOS guideline does not intricate the limitations regarding number of words or pages for chapters/headings provided that the research gap, objectives and significance are clearly mentioned (https://journals.plos.org/plosone/s/file?id=ba62/PLOSOne_formatting_sample_title_authors_affiliations.pdf).

▪Sorrily, we have clearly mentioned the BF and BFI, their theoretical and empirical gaps, including in Nepalese contextual practices as… 

“Breastfeeding (BF) promotion is the most cost-effective intervention to advance mother-child health(1). BF, including early initiation of it (EIBF), not only reduces infant and under-five mortalities but also plays key roles in maternal health promotion by protecting mothers against postpartum bleeding, breast and ovarian cancer, and osteoporosis (2,3). BF should be initiated as early as and beneficial when continued as long as possible, for which a mother should be prepared before a child is delivered, when only a breastfeeding intent (BFI), not the actual behavior, can be assessed. Even more importantly, mothers, mainly nulliparous, do not adhere to BF, mainly EIBF(4). In the same predicament, only 80% of newborn babies were found to be breastfed within one hour of delivery in the Nepalese context (5).”….

▪Kindly, we wanted to clarify in the study that we were more concerned towards the cognitive and social process (social psychological) variables of BFI and more delineated towards this. 

Q

Methods section also needs more detail explanation- research context, selection of participants, variables, data analysis model, results section reported beta coefficient but that was not described in the methods section. Very importantly, there is no data how four constructs- BFA, PBFC, BFSI, and BFSN were created (as these were presented quantitatively)

RE

▪We are thankful for this comment. We added the context, selection of pregnant mothers who completed their first trimesters. 

▪ Thnak you for the comment. For this, we added a Supplemental file (S1. Table 1-16_Constructs and measurement). We also added in the manuscript, with a separate sub-heading, Scoring of constructs, and added text “We assessed BFA with 12 statements, BFSN with two, PBFC with three, BFI with five, BFSI with four, BFK with seven, and BFDN and BFMN each with three statements. All the statements were assessed in 5-point Likert scales, except for BFK and BFDN, with possible value ranges of 12-60, 2-10, 3-15, 5-25, 4-20, and 3-15 for BFA, BFSN, PBFC, BFI, BFSI, and BFMN, respectively. BFK had seven statements with value ranges of 0-1 and 1-3 (possible range; 2-11), whereas BFDN had three statements with value ranges of 0-3, 1-3 and 1-5. Possible values of both BFK and BFDN were 2-11. We dummied the variables with median values for modelling (S1.Table 1-16).”

▪ Kindly would like to reiterate that we have mentioned the stepwise logistic regression model and various related parameters.

Q

Table can be included in the findings section- that reads easy with the text

RE

Thank you for this comment, will be managed further. 

Q

Discussion section lacks contextualities of findings relating with policy and programs of Nepal. Interpretation should be

focused comparing with other studies in Nepal.

RE

Dear Reviewer, thank you for the comment. We added five more recent refs (including in limitations), after which there are 23 refs in discussion, which clearly shows the direct and mediating effects of theoretical constructs including the additional-BFSI and income, we have enriched the discussion. Since this is more a theory testing the mediating effects of BFSI and income, in all the theoretical constructs of TPB in breastfeeding (as we have already mentioned also) rather than directly related with a policy or program, and contextualizing. The study findings would possibly guide the policy or program. If you want to make it differently with policy or program to be related, we request to further clearance in this regard. Moreover, since this is a theory-driven research, and local and contextualized researches are very scarce in Nepalese context. 

REVIEWER #2

Q

It has an ethical statement clearly stated

RE

Thank you for your complement. 

Q

It is better if the document adds operational definitions and discusses all possible reasons for an association.

RE

Thanks for this comment. We have added with a separate sub-heading under “Scoring of constructs” and added the text, as mentioned above (Reviewer #1, Comment 4)

---

## [Editor Report · Decision Letter 1]

9 Jul 2024

PONE-D-23-04076R1Self-identity explains better breastfeeding intention of ethnic pregnant mothers in western Nepal: Extending the theory of planned behaviorPLOS ONE

Dear Dr. Adhikari,

Thank you for submitting your manuscript to PLOS ONE. After careful consideration, we feel that it has merit but does not fully meet PLOS ONE’s publication criteria as it currently stands. Therefore, we invite you to submit a revised version of the manuscript that addresses the points raised during the review process.

I have reviewed your responses to the reviewers. Some responses were not adequate. Please see my requests and provide responses to them.

We look forward to receiving your revised manuscript.

Kind regards,

Jennifer Yourkavitch

Academic Editor

PLOS ONE

Journal Requirements:

Additional Editor Comments:

Thank you for this re-submission. This manuscript needs a thorough editing for English grammar. In addition, please address the following points (from the academic editor):

1. As the reviewer suggested, please develop the Introduction more. You can create at least three paragraphs from what you currently have to address: 1) the topic of predictors of BFI and related literature; 2) BFI in Nepal and especially among the populations of interest (ethnic and low-income), citing related literature; and 3) the purpose of this manuscript.

2. Methods: how did you choose the facilities for this study? Please explain.

3. Did you enroll all pregnant women who presented after their first trimester in the timeframe Dec 2018 - Nov 2019? How were the specific participants enrolled?

4. Please describe more about the sample size calculation. What is p? Is it the prevalence of pregnant women after the first trimester who have a BFI?

5. For scoring the constructs, were standard tools used? If so, please provide references. If not, please justify why standard measures were not used. It looks like you determined "good" and "bad" based on the median scores from this sample. Please justify that approach. It's not clear why the median of this sample would be used to make that judgment.

6. To determine predictors or potential confounders to include in multivariable models, typically a higher p-value is used during bivariate analysis (e.g., p<0.20). Have you considered that approach?

7. The Discussion needs more development. For example, in the last paragraph (before Limitations), you should discuss how your findings were similar or different from the studies you cited.

8. Tables: Define joint and nuclear family in Table 1.

9. Figure 1--you identify betas but I think those values are the aORs.

---

## [Author Response · Author response to Decision Letter 1]

28 Aug 2024

1. As the reviewer suggested, please develop the Introduction more. You can create at least three paragraphs from what you currently have to address: 1) the topic of predictors of BFI and related literature; 2) BFI in Nepal and especially among the populations of interest (ethnic and low-income), citing related literature; and 3) the purpose of this manuscript.

Response: Thank you! We included 15 more references (now 29, including 14 previous) and updated the chapter —Introduction—with three different paragraphs, considering your valuable suggestions:

i) Theoretical constructs and influence on BFI,

ii) Other factors including the self-identity, moral and descriptive norms, health belief, and the sociodemographic factors; and 

iii) BFI in Nepal and the paucity of evidence regarding BFI, and the purpose of the manuscript.

Comment 2:How did you choose the facilities for this study? Please explain.

Res: We selected six health care facilities (HCFs) (Kaski-2, Tanahu-4) in Gandaki province. We selected the health facilities purposively considering higher the attendance of ethnic pregnant mothers. Obviously this may limit the generalizability and so, mentioned in the limitations, too. 

Comment3:Did you enroll all pregnant women who presented after their first trimester in the timeframe Dec 2018 - Nov 2019? How

were the specific participants enrolled?

Res: We appreciate this important question. 

In Nepal, the majority of pregnant women (85.8%) receive the first ANC services within four months of pregnancy. As mothers come generally to the HCFs several weeks after their amenorrhea, it is difficult to get enrolled before their first trimester.

we enrolled conveniently among from those who completed their first trimester. This has been described briefly in methods section. 

Comment 4: Please describe more about the sample size calculation. What is p? Is it the prevalence of pregnant women after the

first trimester who have a BFI?

Res: Yes, the 'p' is the prevalence (proportion) of high intent of mothers for exclusive breastfeeding (EBF)—29.8%. This gives the high intention of mothers to breastfeed in Odissa State of India. Moreover, among the BF indicators, EBF is found with lowest proportion in Nepal, and that was 70% in 2011. Coincidently, these (both) data give a similar size of sample to obtain. 

Comment 5: For scoring the constructs, were standard tools used? If so, please provide references. If not, please justify why

standard measures were not used. It looks like you determined "good" and "bad" based on the median scores from this

sample. Please justify that approach. It's not clear why the median of this sample would be used to make that judgment.

Res: ▪Most of the tools of knowledge, constructs of TPB, and extended variables were adapted and modified according to the Nepalese context, this was pretested, however, with a low sample size, and revised in the final tool. Researchers reviewed the tools and some of the researchers of the team were expert in this field. 

(Pls refer to Supplemental file (S1. Table 1-16_Constructs and measurement). We also described the measurement and scoring of the constructs in the manuscript, with a separate sub-heading, "Scoring of constructs". 

▪ We applied the median cut-offs for fair and poor, positive and negative, and other categorization/dummy of variables, although it is less robust, we applied for multiple logistic regression arbitrarily, and also mentioned in limitations. We also checked for multicollinearity with VIF and Tolerance, which showed no multicollinearity (values of VIF and tolerance given below). 

Variables Collinearity statistics

 Tolerance VIF

Breast Feeding Behavioral Attitude .917 1.091

Subjective Norm .899 1.113

Perceived Behavioural Control .956 1.046

Descriptive Norm .889 1.125

Moral Norm .935 1.070

Self'-Identity .953 1.049

Parity .950 1.053

Annual Income in $US .973 1.028

Comment 6: To determine predictors or potential confounders to include in multivariable models, typically a higher p-value is used

during bivariate analysis (e.g., p<0.20). Have you considered that approach?

Res: ▪We are thankful for your concern. However, having more variables in explanatory categories, we pick-up with a p-value of maximum of 0.06, so that most of the variables could be filtered for stepwise LR. We believe this didn’t miss the important variables. 

Comment 7: The Discussion needs more development. For example, in the last paragraph (before Limitations), you should discuss

how your findings were similar or different from the studies you cited.

Res: Thank you for the suggestion. We have added and covered the issues you have raised. We added three more paragraphs with 22 citations and references further added, totaling of 52 references (from 30 earlier). We also checked and revised discussion. (Kindly refer to track-changed file for reference)

Comment 8: Tables: Define joint and nuclear family in Table 1. 

Res: Thank you for this comment. Added accordingly in the table legend. 

Comment 9: Figure 1--you identify betas but I think those values are the aORs 

Res: Thank you for your complement. Both β's and aORs maintained. Also the figures are slightly modified.

---

## [Editor Report · Decision Letter 2]

4 Sep 2024

PONE-D-23-04076R2Self-identity explains better breastfeeding intention of ethnic pregnant mothers in western Nepal: Extending the theory of planned behaviorPLOS ONE

Dear Dr. Adhikari,

Thank you for submitting your manuscript to PLOS ONE. After careful consideration, we feel that it has merit but does not fully meet PLOS ONE’s publication criteria as it currently stands. Therefore, we invite you to submit a revised version of the manuscript that addresses the points raised during the review process.

We look forward to receiving your revised manuscript.

Kind regards,

Jennifer Yourkavitch

Academic Editor

PLOS ONE

Journal Requirements:

Additional Editor Comments:

Thank you! Please check again for correct English grammar throughout the manuscript. For example, in the Methods section of the Abstract, you want to say "six purposively selected..." rather than the current phrasing. In the Methods section, please spell out all of the acronyms at the first usage (under "Scoring of Constructs"). Please also put the footnote about family type in Table 1, where the distinction first appears.

---

## [Author Response · Author response to Decision Letter 2]

25 Sep 2024

Dear Editor, 

thank you for your inputs

Here, we keep the comments and the responses:

1. Journal requirement 

Response

Thank you, for making aware about the journal requirement. We checked all the references, 1-52, for completeness. For retraction check, we visited The Retraction Watch Database (http://retractiondatabase.org/RetractionSearch.aspx?AspxAutoDetectCookieSupport=1#?AspxAutoDetectCookieSupport%3d1%26ttl%3dInfant%2bfeeding%2binformation%252c%2battitudes%2band%2bpractices%253a%2ba%2blongitudinal%2bsurvey%2bin%2bcentral%2bNepal) mainly, and also in the “PUBPEER” in some selected cases, and in some, when the ‘Watch database’ was not giving clear retraction. 

 We did not find any of our references ‘retracted’. 

2. Whole manuscript 

Please check again for correct English grammar throughout the manuscript. For example, in the Methods section of the Abstract, you want to say "six purposively selected..." rather than the current phrasing.

Response

Thank you for this point. We corrected this one, and again, for the whole document, we checked in online platform, for grammar check (https://quillbot.com/grammar-check), by each sentence. We also checked manually, line-by-line for the errors. 

3. Methods

Spell out all of the acronyms at the first usage (under "Scoring of Constructs"). Please also put the footnote about family type in Table 1, where the distinction first appears.

Response

Thank you for the comment!

We have done with these all. 

women (85.8%) receive the first ANC services within four months of pregnancy. As mothers come generally to the HCFs several weeks after their amenorrhea, it is difficult to get enrolled before their first trimester.

we enrolled conveniently among from those who completed their first trimester. This has been described briefly in methods section. 

4. Methods

Also put the footnote about family type in Table 1, where the distinction first appears.

Response

Sorry for this typo error, as earlier it was supposed to be done. Now addressed, and checked once again. 

5.Figures 

Checking figures in the “PACE” platform for journal compatibilities.

Response

Thank you! We checked without any difficulty, and as suggested, we are now uploading the ‘produced’ ones. 

Comments on 24 and 25th Sep from Journal office

1.Files 

We note that several of your files are duplicated on your submission. Please remove any unnecessary or old files from your revision, and make sure that only those relevant to the current version of the manuscript are included. 

Response

Thank you. I have removed the older correction/trackchanged and other files, and tetained the recent files. 

2. Title 

Please amend the title either on the online submission form or in your so that they are identical. 

Response

We are extremely sorry, to update in cover letter, now checked and corrected in online, manuscript and other submitted documents. It is consistent now as "“Self-identity explains better breastfeeding intention of ethnic pregnant mothers of Western Nepal: Extending the theory of planned behavior”. Kindly let us remind if still there is any discrepancy. 

3. Figure 

Please remove your figures/ from within your manuscript file, leaving only the individual TIFF/EPS image files. Please include a separate legend for each figure in your manuscript. 

Response

The figures that have been 'produced' from the PACE have been uploaded, and the legend has been been separated from the main figure and kept where they occur in the text (manuscript). 

Thank you for your inputs, comments, and scientific contributions!

---

## [Editor Report · Decision Letter 3]

30 Sep 2024

Self-identity explains better breastfeeding intention of ethnic pregnant mothers of Western Nepal: Extending the theory of planned behavior

PONE-D-23-04076R3

Dear Dr. Adhikari,

We’re pleased to inform you that your manuscript has been judged scientifically suitable for publication and will be formally accepted for publication once it meets all outstanding technical requirements.

Kind regards,

Jennifer Yourkavitch

Academic Editor

PLOS ONE
---

## [Editor Report · Acceptance letter]

4 Oct 2024

PONE-D-23-04076R3 

PLOS ONE

Dear Dr. Adhikari, 

I'm pleased to inform you that your manuscript has been deemed suitable for publication in PLOS ONE. Congratulations! Your manuscript is now being handed over to our production team.

Kind regards, 

on behalf of

Dr. Jennifer Yourkavitch 

Academic Editor

PLOS ONE